# The CDK Inhibitor Dinaciclib Improves Cisplatin Response in Nonseminomatous Testicular Cancer: A Preclinical Study

**DOI:** 10.3390/cells13050368

**Published:** 2024-02-20

**Authors:** Elisa Rossini, Mariangela Tamburello, Andrea Abate, Silvia Zini, Giovanni Ribaudo, Alessandra Gianoncelli, Stefano Calza, Francesca Valcamonico, Nazareno R. Suardi, Giuseppe Mirabella, Alfredo Berruti, Sandra Sigala

**Affiliations:** 1Section of Pharmacology, Department of Molecular and Translational Medicine, University of Brescia, 25123 Brescia, Italy; elisa.rossini@unibs.it (E.R.); andrea.abate@unibs.it (A.A.); s.zini004@unibs.it (S.Z.); giovanni.ribaudo@unibs.it (G.R.); alessandra.gianoncelli@unibs.it (A.G.); sandra.sigala@unibs.it (S.S.); 2Unit of Biostatistics and Bioinformatics, Department of Molecular and Translational Medicine, University of Brescia, 25123 Brescia, Italy; stefano.calza@unibs.it; 3Oncology Unit, Department of Medical and Surgical Specialties, Radiological Sciences, and Public Health, University of Brescia at ASST Spedali Civili di Brescia, 25123 Brescia, Italy; franzval@yahoo.it (F.V.); alfredo.berruti@gmail.com (A.B.); 4Urology Unit, Department of Medical and Surgical Specialties, Radiological Sciences, and Public Health, University of Brescia at ASST Spedali Civili di Brescia, 25123 Brescia, Italy; nazareno.suardi@unibs.it (N.R.S.); giuseppe.mirabella@unibs.it (G.M.)

**Keywords:** testicular cancer, cisplatin resistance, CDK inhibitors, dinaciclib, combined treatment, zebrafish xenograft

## Abstract

Background: Most patients with testicular germ cell tumors (GCTs) are treated with cisplatin (CP)-based chemotherapy. However, some of them may develop CP resistance and therefore represent a clinical challenge. Cyclin-dependent kinase 5 (CDK5) is involved in chemotherapy resistance in different types of cancer. Here, we investigated the possible role of CDK5 and other CDKs targeted by dinaciclib in nonseminoma cell models (both CP-sensitive and CP-resistant), evaluating the potential of the CDK inhibitor dinaciclib as a single/combined agent for the treatment of advanced/metastatic testicular cancer (TC). Methods: The effects of dinaciclib and CP on sensitive and resistant NT2/D1 and NCCIT cell viability and proliferation were evaluated using MTT assays and direct count methods. Flow cytometry cell-cycle analysis was performed. The protein expression was assessed via Western blotting. The in vivo experiments were conducted in zebrafish embryos xenografted with TC cells. Results: Among all the CDKs analyzed, CDK5 protein expression was significantly higher in CP-resistant models. Dinaciclib reduced the cell viability and proliferation in each cell model, inducing changes in cell-cycle distribution. In drug combination experiments, dinaciclib enhances the CP effect both in vitro and in the zebrafish model. Conclusions: Dinaciclib, when combined with CP, could be useful for improving nonseminoma TC response to CP.

## 1. Introduction

Among germ cell tumors (GCTs), testicular cancer (TC) is the most common type of cancer in young men [1], and the high rate of heterogeneity distinguishes it from other solid tumors [2]. Indeed, GCTs are divided into seminomas and nonseminomas, displaying significant differences in terms of both treatment options and response to therapy [3]. Seminomas are homogeneous cancers of embryonic germ cells, while nonseminomas may contain one or more histological subtypes, including embryonal carcinomas, yolk sac tumors, choriocarcinomas and teratomas, among others [4]. Nowadays, the standard treatment of GCTs involves orchiectomy followed by cisplatin (CP)-based chemotherapy and/or radiotherapy, according to cancer histology and disease stage. The cure rate is up to 90%. However, 15–20% of patients develop disseminated disease relapse, and this is more common in patients with nonseminomatous GCTs (3.2%) than in patients with seminomatous GCTs (1.4%) [5,6]. Salvage therapy with surgery and standard/high-dose chemotherapy with autologous stem cell transplantation could be effective in around 50% of these patients [7], but no alternative treatment options are available for TC patients who do not respond to these regimens. CP resistance is among the main causes of treatment failure and progression in patients with TC, but its biological background is still poorly understood [8]. CP resistance seems to be a multifactorial phenomenon that includes upregulation of DNA damage response (DDR) pathways [9,10]. Recently, the role of CDKs in DDR has emerged. CDKs affect both damage signaling and DNA repair, contributing to the fidelity of the cell division process as well as the maintenance of genomic integrity following DNA damage. This is due to the modulatory role of CDKs in double-strand break repair components, including their influence on enzymes involved in homologous recombination (HR) and nonhomologous end-joining (NHEJ) [11]. Of these CDKs, CDK5, an atypical cyclin-dependent kinase, takes part in the DDR process mainly by phosphorylating some of the critical DDR proteins such as ataxia–telangiectasia mutated (ATM) and apurinic/apyrimidinic endonuclease 1 (Ape1) [12]. Accordingly, it has been demonstrated that in HeLa cells overexpressing cyclin I, the activation of CDK5 by cyclin I confers cancer cell resistance to CP, while knockdown of CDK5 with siRNA significantly increases the sensitivity to CP [13]. The involvement of this kinase in the establishment of CP resistance could be hypothesized based on these observations.

This hypothesis is further supported by results reported in CDK5-inhibited HCC cells [14] and CDK5-depleted ovarian cancer cell lines [15], in which cancer cells exhibit higher sensitivity to DNA-damaging agents. Thus, we investigated the possible role of CDK5 in influencing the CP effects in nonseminoma TC cell models based on a shared approach for the treatment of metastatic disease, the need to bypass CP resistance and the emerging role of CDKs in TC [16,17]. Among CDK5 inhibitors [18], we focused our attention on dinaciclib, a potent and selective small molecule inhibitor of CDK2, CDK5, CDK1 and CDK9, since the 50% inhibitory concentration values are in the nM range [19,20] and it is currently in a phase III clinical trial (NCT01580228) [21]. We evaluated the potential of dinaciclib as a new pharmacological approach alone or in combination with CP for the treatment of advanced/metastatic nonseminomas. This hypothesis is supported by the finding that in preclinical cancer models such as ovarian cancer cell lines, dinaciclib synergizes with CP in killing cancer cells [22].

## 2. Materials and Methods

### 2.1. Cell Lines

The NT2/D1 (ATCC CRL-1973) cell line was purchased from American Type Culture Collection (ATCC, Manassas, VA, USA) and cultured as indicated by the manufacturer. The NCCIT cell line and the CP resistant subclones, namely NT2/D1–R and NCCIT–R, were kindly provided by Prof. Bremmer (Gottingen, Germany) and cultured as suggested [23]. According to previously published data, NT2/D1-R and NCCIT-R cell lines were maintained and periodically confirmed as resistant to CP [23,24]. Media and supplements were supplied by Merck KGaA (Darmstadt, Germany). Cell lines were periodically both authenticated (BMR Genomics, Padova, Italy) and tested for mycoplasma.

### 2.2. Cell Viability and Cell Proliferation Assay

Cells (10,000/well) were seeded in a 24-well plate. Twenty-four hours later, cells were treated with increasing concentrations of CP (0.05 µM–15 µM) or dinaciclib (0.1 nM–15 nM) for 48 h, according to the calculated doubling time [25]. CP and dinaciclib were purchased from Selleck Chemicals (Milan, Italy) and solubilized in dimethylformamide and DMSO, respectively. Cell viability was assessed using 3-(4,5 dimethyl-2-thiazol)-2,5-di-phenyl-2htetrazolium bromide (MTT) dye reduction assay, as described [26]. Cell proliferation was evaluated by direct counting using a MACSQuant10 Analyzer 

(Miltenyi Biotec GmbH, Bielefeld, Germany). Briefly, cells (80,000/well) were seeded in a 6-well plate. Twenty-four hours later, cells were treated with IC_25_, IC_50_ and IC_80_ concentrations of CP or dinaciclib. Forty-eight hours later, cells were trypsinized and counted using aMACSQuant10 Analyzer (Miltenyi Biotec GmbH, Bielefeld, Germany). Data were analyzed using FlowJo v10.6.2 software(TreeStar, Ashland, OR, USA).

### 2.3. Combined Drug Treatment: CP plus Dinaciclib

To study the effect of the combined treatment of CP plus dinaciclib on nonseminoma cell viability, cells (10,000/well) were seeded in a 24-well plate and 24 h later were treated as described with increasing concentrations of CP (0.05 µM–15 µM) in the presence/absence of dinaciclib IC_25_/IC_50_ concentrations previously obtained. Forty-eight hours after treatment, the MTT assay was performed as described above. The cell proliferation rate was evaluated by seeding cells (1.5 × 105/well) in a 6-well plate and treating them with IC_25_, IC_50_ and IC_80_ concentrations of CP in the presence of dinaciclib IC_25_/IC_50_ concentrations. Forty-eight hours after treatment, cells were trypsinized and counted using aMACSQuant10 Analyzer (Miltenyi Biotec GmbH, Bielefeld, Germany). Data were analyzed using FlowJo v10.6.2 software(TreeStar, Ashland, OR, USA).

### 2.4. Cell-Cycle Analysis

To verify whether CP and dinaciclib exposure interferes with nonseminoma cell-cycle distribution, 10^6^ cells/10 mm dish were plated and treated for 48 h with the IC_50_ concentration of CP/dinaciclib alone and in combination. At the end of the treatment period, cells were harvested and the cell-cycle distribution was analyzed as described [27] using the MACSQuant10 Analyzer (Miltenyi Biotec GmbH, Bielefeld, Germany). Data were analyzed using FlowJo v10.6.2(TreeStar Inc., Ashland, OR, USA).

### 2.5. Tumor Xenograft

Zebrafish were maintained and used according to EU Directive 2010/63/EU for animal use following protocols approved by the local committee (OPBA) and authorized by the Ministry of Health (Authorization Number 393/2017). Adult transgenic line Tg (kdrl:EGFP) and wild-type zebrafish lines were maintained as described in [28]. To evaluate the toxic effect of dinaciclib on the zebrafish model, 48 hpf wild-type (wt) embryos (AB) were divided into different groups as indicated and maintained in PTU/fish water, to which solvent (DMSO) and 0.025, 0.05, 0.1 or 0.5 μM dinaciclib were added. After 3 days (T3), the drug effects were observed. To evaluate the effect of dinaciclib on tumor growth, Tg (kdrl:EGFP) zebrafish embryos at 48 hpf were dechorionated, anesthetized with 0.042 mg/mL tricaine and microinjected with labeled CP-sensitive/resistant NT2/D1 and CP-sensitive/resistant NCCIT cells into the subperidermal space of the yolk sac as described [29,30]. Approximately 250 cells/4 nL were injected into each embryo (about 50 embryos/group), and embryos were maintained in PTU/fish water in a 32 °C incubator to allow tumor cell growth. Pictures of injected embryos were acquired 2 h after cell injection (T0) using a Zeiss Axio Zoom.V16 (Zeiss, Jena, Germany) fluorescence microscope, equipped with Zen 2.3 Blu software and PlanNeoFluar Z 1.0X objective. Zebrafish embryos were treated with solvent, 0.1 µM of dinaciclib and 10 µM of CP alone or in combination, adding them directly to the PTU/fish water. After 3 days of treatment (T3), pictures were taken as indicated above. After 3 days (T3), the effects of the drugs on cancer cell growth were recorded by taking pictures to measure the tumor areas of each group at T0 and T3 using Zen 2.3 Black software (ZEISS, Jena, Germany). Some representative xenografted embryos were fixed, embedded in low melting agarose and imaged using an LSM 900 confocal laser microscope equipped with an Achropla 10×/0.25 objective. Dinaciclib absorption from embryos was evaluated by quantifying the concentration of dinaciclib using liquid chromatography–tandem mass spectrometry (LC-MS/MS) as described in the Appendix A.

### 2.6. Western Blot

Cells were lysed in ice-cold RIPA buffer with a complete set of protease and phosphatase inhibitors (Roche Italia, Monza, Italy). The protein concentration was measured using the Bradford Protein Assay, and the same amount of whole lysate was separated using electrophoresis on a 4–12% Bis–Tris gel and electro-blotted to a PVDF membrane, following the manufacturer’s instructions. Membranes were reacted using the primary antibodies shown in Appendix A. Secondary antimouse (IRDye 680CW conjugated) and antirabbit (IRDye 800CW conjugated) antibodies (final concentration: 0.67 µg/mL; LI-COR Biosciences, Lincoln, NE, USA) were applied for 1 h at room temperature. The specific signal was visualized using the Odyssey Imaging System (LI-COR Biosciences), and the densitometry analysis was performed using the Image Studio TM Light V 5.2 Software (LI-COR Biosciences, Lincoln, NE, USA).

### 2.7. Statistical Analysis

Data analysis was conducted using Prism 5.0 (GraphPad Software, San Diego, CA, USA). Statistical analysis was carried out using one-way analysis of variance (ANOVA) and Bonferroni’s multiple comparison test or Student’s *t*-test. *p* < 0.05 was considered statistically significant. Unless otherwise specified, data are expressed as mean ± SEM of at least three experiments run in triplicate.

## 3. Results

### 3.1. Protein Expression of CDKs Targeted by Dinaciclib and Their Related Cyclins

Dinaciclib can bind and inhibit several CDKs, including CDK1, CDK2, CDK5 and CDK9. We evaluated the protein expression of these CDKs and some related cyclins in CP-sensitive/resistant nonseminoma experimental cell models, namely NT2/D1/-R and NCCIT/-R cells (Figure 1). Western blot results revealed that CDK1 was highly expressed in each cell model, with no significant differences between CP-sensitive and CP-resistant cells.

Among the activators of CDK1, cyclin A exhibited higher expression levels (compared to those of the other activator cyclin B2) with a notable trend of increased expression in CP-resistant cells (NT2/D1-R: +370.9%, *p* < 0.001; NCCIT-R: +117.2%, *p* < 0.05). Conversely, CP-resistant cells showed lower levels of cyclin B2 expression (NT2/D1-R: −43.5%, *p* < 0.05; NCCIT-R: −85.0%, *p* < 0.05).

CDK5 protein expression was significantly upregulated in CP-resistant cells compared to the sensitive models (NT2/D1-R: +67.05%, *p* < 0.05; NCCIT-R: +369.37%, *p* < 0.001). However, the levels of CDK5 activators, namely cyclin I, p35 and p39, in terms of both gene expression and protein abundance were uniformly low across all cell models, exhibiting no significant differences (Figure 1, Appendix A). Concerning CDK2, we observed lower protein levels, along with CDK9, whose expression appeared to increase in NT2/D1-R cells (+272.9% vs. NT2/D1 cells, *p* < 0.001). In addition to cyclin A, CDK2 can be activated by cyclin E, which was highly expressed in NT2/D1-R cells (+971.9% vs. NT2/D1 cells, *p* < 0.0001). A similar trend was observed for the activators of CDK9, cyclin T1 and K (cyclin T1 in NT2/D1-R: +481.01%, *p* < 0.0001; cyclin K in NT2/D1-R: +139.6%; cyclin K in NCCIT-R: +162.0%).

### 3.2. Effect of Dinaciclib Treatment on NT2/D1/-R and NCCIT/-R Cells

The CP-sensitive/resistant nonseminoma cell models were treated with increasing concentrations of dinaciclib, as described above. A sigmoidal dose–response function was applied to calculate the IC_50_ values (Table 1). In both CP-sensitive and CP-resistant NT2/D1 (Figure 2-1a) and NCCIT (Figure 2-2a) dinaciclib induced a concentration-dependent reduction in cell viability and cell proliferation rate (Figure 2(1b,2b)). To evaluate whether dinaciclib could affect the cell-cycle distribution, cells were treated with the dinaciclib IC_50_ concentrations and analyzed by flow cytometry. As reported in Figure 2-1c, dinaciclib induced a significant increase in the SUB-G1 phase of the cell cycle in NT2/D1 cells (untreated: 0.44% ± 0.17%, dinaciclib-treated: 1.29% ± 0.18%; *p* < 0.05). In NCCIT cells, we observed a significant increase in the percentage of cells in SUB-G1 and G2 phases after dinaciclib treatment (untreated, SUB-G1: 0.65% ± 0.13%, G2: 21.12% ± 0.68%; dinaciclib-treated: SUB-G1: 2.93% ± 0.48%, G2: 25.49 ± 1.02%; *p* < 0.01). Interestingly, no significant alterations in cell-cycle distribution were observed in the CP-resistant clones. The increase in the SUB-G1 fraction after dinaciclib treatment could be due to DNA fragmentation, suggesting that apoptosis could be the mechanism mediating drug cytotoxicity. To verify this hypothesis, the annexin/PI assay was performed. We observed a dinaciclib-induced increase in apoptotic cells in both NT2/D1 cells (untreated: 7.05% ± 3.26%, dinaciclib-treated: 72.20% ± 7.90%; *p* = 0.008) and NCCIT cells (untreated: 7.25% ± 3.13%, dinaciclib-treated: 16.82% ± 0.43; *p* = 0.03). These data confirm the involvement of apoptosis in the dinaciclib cytotoxicity. A representative image is reported in Appendix A.

### 3.3. Effect of Combined Treatment of Dinaciclib/CP in NT2/D1 and NCCIT-Sensitive and CP-Resistant Cells

The effect of CP was first evaluated in all cell models, exposing them to increasing concentrations of the drug. Cell viability was evaluated, and the concentration–response curves confirmed different sensitivity profiles among the cell types. The sigmoidal dose–response function was applied, and the IC_50_ values were calculated: NT2/D1 = 0.80 µM (95% CI: 0.44–1.45); NT2/D1-R = 4.22 µM (95% CI: 2.96–6.01); NCCIT = 3.70 µM (95% CI: 2.73–5.03) and NCCIT-R = 5.39 µM (95% CI: 2.73–10.63). The calculated values are consistent with the published results [25]. The effect on cell viability of CP and dinaciclib in combination settings was then analyzed in both sensitive and CP-resistant cell lines. Cells were treated with increasing concentrations of CP alone or combined with the IC25 or IC50 concentrations of dinaciclib. Results of concentration–response curves are reported in Figure 3-1/2a (NT2/D1-sensitive/R) and Figure 4-1/2a (NCCIT-sensitive/R). Efficacy was chosen as the pharmacological parameter to compare CP alone and CP plus dinaciclib treatments from a functional point of view (Table 2). Significantly greater efficacy for combined treatment (both with the IC_25_ and IC_50_ of dinaciclib) than for treatment with CP alone was reported in CP-resistant models, while nonsignificant modifications were observed in CP-sensitive models. We have previously demonstrated the CP-related increase in NT2/D1-sensitive cells in the G2 phase [17]. Here, we also investigated the effect of CP plus dinaciclib treatment on cell-cycle distribution in this cell model. No significant modification in the percentage of cells in each cell-cycle phase was observed after the combined treatment (Figure 3-1b and Appendix A). In the resistant NT2/D1-R subclone, CP alone induced an increased G2 phase (untreated, G2: 19.85% ± 4.18%, CP-treated, G2: 76.83% ± 10.49%; *p* < 0.001) and significantly decreased G1 and S fractions (untreated, G1: 51.08% ± 6.72%, S: 25.73% ± 1.72%; CP-treated, G1: 12.19% ± 6.51%, S: 6.18% ± 3.28%; *p* < 0.01, *p* < 0.001). These effects were maintained if dinaciclib and CP were combined (Figure 3-2b). In NCCIT-sensitive cells, we reported an increase in SUB-G1, G1 and G2 fractions and a decrease in S fraction after treatment with CP alone (untreated, SUB-G1: 0.96% ± 0.25%, G1: 42.25% ± 2.42%, S: 32.48% ± 4.49%, G2: 24.00% ± 2.45%; CP-treated, SUB-G1: 5.97% ± 0.94%, G1: 50.01% ± 0.10%, S: 1.26% ± 1.09%, G2: 43.09% ± 0.48%; *p* < 0.01, *p* <0.05, *p* < 0.001, *p* < 0.001). When CP was combined with dinaciclib, the same trend was observed, but the statistical significance was reached only for the S phase reduction (Figure 4-1b). The increase in NCCIT SUB-G1 cells after CP treatment suggests the induction of the apoptotic mechanism, confirmed by the annexin/PI assay (Appendix A). Indeed, NCCIT apoptotic cells was 5.59% ± 0.74% in the untreated condition and 22.54% ± 1.40% in CP-treated cells (*p* = 0.0002). Concerning NCCIT-R cells, we observed an increased G2 fraction and a decreased S fraction after CP treatment (untreated, S: 44.15% ± 3.80%, G2: 23.06% ± 5.53%; CP-treated, S: 1.53% ± 1.36%, G2: 57.08 ± 4.04%; *p* < 0.0001, *p* < 0.01). A nonsignificant increase in the G1 fraction was also reported. The effect was maintained in the combined treatment (Figure 4-2b).

### 3.4. Effect of Dinaciclib Alone or Combined with Cisplatin in the Zebrafish/Tumor Xenograft Model

We next evaluated whether the cytotoxic effect induced in vitro by dinaciclib either alone or in combination with CP also occurs in in vivo experimental models, represented by nonseminoma cells xenografted in kdrl-GFP zebrafish embryos, a well-established model for drug screening. Preliminary experiments were conducted to evaluate the dinaciclib toxicity in wild-type (AB) zebrafish embryos by exposing 48-hpf zebrafish embryos to increasing concentrations of dinaciclib up to 0.5 µM. After three days (T3), no toxic effects or side effects such as pericardial edema, yolk sac edema, spinal deformity, and mortality were observed even at the highest dose. Consequently, the concentration of 0.1 µM dinaciclib was selected to study its effects on CP-sensitive/resistant NT2/D1 and CP-sensitive/resistant NCCIT cell growth in the zebrafish model. As regards the CP concentration, we based our choice on our previous findings [17], and a dose of 10 µM of CP was used.

Results obtained, in terms of tumor area, are shown in Figure 5 and Table 3 for each experimental group, which received the vehicle, 0.1 µM dinaciclib, 10 µM CP, or a combination of 0.1 µM dinaciclib and 10 µM CP. Dinaciclib caused a significant reduction in terms of tumor area in each xenografted cell model. Its effect was enhanced by cisplatin in embryos xenografted with NT2/D1-R, NCCIT and NCCIT-R cells, although it did not reach the statistical significance required to be defined as additive/synergic. In NT2/D1 the combination of cisplatin and dinaciclib seemed to induce a functional antagonism (Appendix A). Moreover, dinaciclib absorption from embryos was evaluated by quantifying the concentration of dinaciclib using liquid chromatography–tandem mass spectrometry (LC-MS/MS) as described in the Appendix A. The concentration of dinaciclib was analyzed at three time points: at 24 h, it was 0.283 nM ± 0.071; at 48 h, it was 0.151 nM ± 0.005 and at 72 h, the concentration of dinaciclib was lower than the limit of quantification.

## 4. Discussion

CP has proven to be a crucial component for the treatment of TC [31]; indeed, it contributed to the increase in the overall 5-year survival rate from 5% to 80% [32]. Although first-line therapy for the treatment of TC is well defined in accordance with the guidelines of the International Germ Cell Cancer Cooperative Group (IGCCCG), there is still no well-defined protocol for the treatment of chemotherapy-resistant disease. The mechanisms of CP resistance are multifactorial, suggesting the absence of a uniform cause [10]; among these, CDK5 seems to be one of the factors involved in the resistance to DNA alkylating agents [33]. Promising results have emerged from preclinical research supporting the potential involvement of CDKs and CDK inhibitors in triggering cytotoxicity and cell death in TC cell lines [16]. Here, we investigated the effect of the potent CDK inhibitor dinaciclib on nonseminoma GCTs based on this evidence.

Firstly, we quantified the protein expression levels of specific CDKs targeted by dinaciclib, along with their associated cyclins. This analysis aimed to discern any distinctions between sensitive and CP-resistant subclones, ultimately facilitating the identification of the key CDK(s) responsible for mediating the effects of dinaciclib. Interestingly, CDK5 was significantly overexpressed in both CP-resistant cell lines, supporting the hypothesis of an involvement of its expression in mediating CP resistance in TC. CDK5 is known for its role in the regulation of neuronal functions, but its presence and activity in other tissues have also been established [34]. In mice, CDK5 has been detected in the cytoplasm of Sertoli cells and in metaphase spermatocytes [35], and it has been identified in mouse Leydig TM3 and Sertoli TM4 cell lines [36]. Moreover, p35-associated CDK5 activity was observed in rat testis [37]. In this connection, proteomic studies in humans showed that the testis is the organ in which CDK5 is more abundant besides the central nervous system, where the protein is mainly located in the cerebral cortex, cerebellum, hippocampus, and caudate. CDK5 is reported as not prognostic in testis cancer, and the expression of its RNA indicates low cancer specificity. Nevertheless, protein expression data show that 9 of 12 patients with testis cancer show high/medium expression of CDK5 (Human Protein Atlas, proteinatlas.org, accessed on 12 January 2024) [38].

Looking at other members of the CDK family, significantly higher levels of CDK9, together with cyclin T1, were expressed in NT2/D1-R compared to wild-type cells. A trend to increase in CP-resistant cell lines was observed in cyclin K. Intriguingly, insight into the role of CDK9–cyclin K emerged with the identification of cyclin K as a transcription target for p53 in response to DNA damage [39]. Indeed, loss of CDK9 activity causes an increase in spontaneous levels of DNA damage in replicating cells and a decreased ability to recover from a transient replication arrest. This activity is restricted to CDK9–cyclin K complexes and is independent of the CDK9–cyclin T complex [40,41]. Additionally, our model NT2/D1-R significantly overexpresses a CDK2 activator, cyclin E, in line with data indicating that cyclin E overexpression and CDK2 hyperactivation were observed in the CP-resistant lines [42,43]. CDK2 interacts with a large number of proteins that are involved in key cellular processes such as DNA replication, cell-cycle progression, DNA repair and chromatin modeling, thus suggesting a crucial role for CDK2 in orchestrating a fine balance between cellular proliferation, cell death and DNA repair [43,44]. Downregulation of CDK2 can induce sustained DNA damage and elicit the DDR in human embryonic stem cells causing apoptosis [44]. Moreover, it appears that the availability of CDK2 is essential for the DNA repair function of cyclin A1. As a result, cyclin A1 fails to participate in DNA repair in the absence of CDK2 [45,46]. Our data show that cyclin A is significantly overexpressed in CP-resistant subclones. Increased expression of cyclin A and dysregulation of cell-cycle checkpoints promote proliferation of cancer cells, which can be facilitated by phosphorylation of oncoproteins and tumor suppressors [47]. Overexpression of cyclin A has been discovered in human cancer, including ovarian tumors, and it was reported that cyclin A may be used as a biomarker to predict the response to chemotherapy in patients with cancers of different origin [48]. Finally, it has been reported that that compromised CDK1 activity dramatically increases the efficacy of chemotherapeutic acting on DNA replication [49]. Inhibition of CDK1 synergized with cisplatin to induce mitotic cell death in p53-deficent cells and overcame cisplatin resistance in small cell lung cancer preclinical models in vitro and in vivo [50].

Collectively, our Western blot analyses demonstrate that the CDKs targeted by dinaciclib are expressed in the used cell models, showing a consistent increase in their expression along with their associated cyclins in CP-resistant subclones. These findings support the potential application of dinaciclib in TC therapy. Notably, CDK5 stands out as the most prominently expressed target in both resistant clones, further confirming its pivotal role in CP resistance and its mediation of dinaciclib’s effects.

Dinaciclib hampers cell viability and cell proliferation at nanomolar ranges in both sensitive and CP-resistant nonseminoma cell models; in particular, our cells were more sensitive to dinaciclib compared to results published in the literature [16], and this could be due to different cell culture conditions. Regardless, the resistance to CP did not correlate with any decrease in sensitivity to dinaciclib, in accordance with published results [51]. The different trend of response to dinaciclib treatment between NT2/D1/-R and NCCIT/-R could be explained by the presence of mutated p53 in the NCCIT cell model and wild-type p53 in NT2 cells [52]. In general, CDK inhibitors are less effective or not at all in p53-intact cells, as DNA damage-induced activation of p53 prevents premature cell-cycle progression, but much more effective in p53-deficient cancer cells at potentiating cytotoxic agents [50]. Additionally, it has been demonstrated in a previous study that the status of p53 correlates with dinaciclib-induced apoptosis [53]. The cell-cycle distribution was modified by CDK inhibition, as we observed an increase in the percentage of NT2/D1 and NCCIT cells in the SUB-G1 phase of the cell cycle, in line with published data [53,54], highlighting the activation of a strong apoptotic response and cytotoxic activity. No alterations in cell-cycle distribution were observed in CP-resistant cells after dinaciclib treatment alone; however, dinaciclib escalated the CP effect, as demonstrated in different cancer cell models [22,51]. In this regard, we observed a significant increase in CP efficacy in NT2/D1–R and NCCIT-R when the two drugs were combined. The in vitro results were strengthened by data obtained from xenografting the nonseminoma cell lines in zebrafish embryos. In particular, exposure to dinaciclib increased the efficacy of CP-induced reduction of tumor area in three of the cell models used. The functional antagonism observed in NT2/D1 cells could find its rationale in the high sensitivity of these cells to both CP and dinaciclib, which, for each drug alone, reached maximum efficacy in the in vivo model. We would like to underline that the low-nM-effective concentrations of dinaciclib used in our experiments are far below the average dinaciclib concentrations in solid tumors, which are reported to be 82.3–184 nM [55,56].

Dinaciclib has been in the developmental stage for a few years, primarily due to its low safety profile and limited efficacy [21]. Reported on the website https://ClinicalTrials.gov as of 30 January 2024, a total of 18 studies investigating the potential efficacy of dinaciclib, either as a single or combined treatment across various cancer types, have been identified. Among them, two are currently ongoing. One is in phase II, assessing the potential impact of dinaciclib on patients with stage IV melanoma, while the other is a phase I study evaluating the safety of combining dinaciclib with the PARP inhibitor veliparib in patients with advanced solid tumors.

Interestingly, the first Phase I clinical trial on dinaciclib as a single agent performed on patients with advanced malignancies reports the achievement of prolonged stable disease for at least four treatment cycles in 10 up to 48 enrolled patients, with mild adverse effects (nausea, anemia, decreased appetite and fatigue) [19].

Furthermore, several phase II trials have investigated dinaciclib [57], including a study that compared dinaciclib to capecitabine in patients with advanced breast cancer [56]. The dinaciclib treatment exhibited antitumor activity in two out of seven patients with ER+ and Her2- metastatic breast cancer, although the efficacy did not surpass that of capecitabine (*p* = 0.991). Common toxicities observed included neutropenia, leukopenia, an increase in aspartate aminotransferase, and febrile neutropenia. Dinaciclib has entered Phase III development for refractory chronic lymphocytic leukemia [58].

In terms of efficacy, it is widely recognized that nonselective CDK inhibitors, including dinaciclib, have limited value as standalone antineoplastic agents when translated to clinical settings [59]. However, recent advancements in selective CDK inhibitors have marked the initial success in targeting these kinases for therapeutic purposes in various diseases. Combination therapies involving CDK inhibitors show greater promise than monotherapies, suggesting the need to evaluate multiple chemotherapeutic agents in conjunction with CDK inhibitors. The effectiveness of other targeted drugs in combination also underscores the importance of further exploration in clinical research.

While we recognize the inherent risks associated with the translation of preclinical findings to clinical outcomes, it is noteworthy that our models indicate dinaciclib’s ability to enhance the effectiveness of CP even at very low nanomolar concentrations.

## 5. Conclusions

Taken together, our findings suggest that dinaciclib, whether administered as a single agent or in combination with CP, effectively reduced the viability and proliferation of nonseminoma TC cell lines, particularly in those that are CP-resistant. This highlights the potential of dinaciclib in addressing CP resistance in nonseminoma TC, emphasizing the utility of in vitro cellular models that mirror the diverse phenotypes encountered in clinical practice.

## Figures and Tables

**Figure 1 cells-13-00368-f001:**
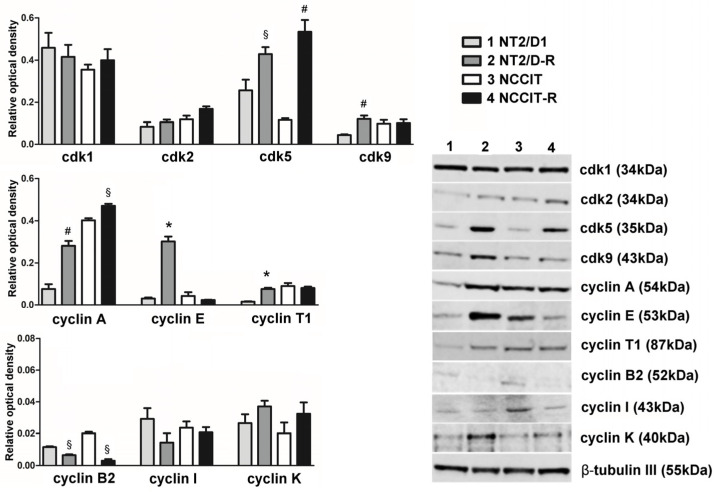
CDKs targeted by dinaciclib and their related cyclin protein expression in NT2/D1, NT2/D1-R, NCCIT and NCCIT-R cells. A total of 30 µg of total cell lysate was separated on a 4–12% Bis–Tris gel as described. Lane 1—NT2/D1, lane 2—NT2/D1-R, lane 3—NCCIT, lane 4—NCCIT-R. The human β-tubulin was used as an internal control. Quantification results are presented as a relative optical density means ± SEM of three independent experiments, and representative Western blot results are shown. * *p* < 0.0001, # *p* < 0.001, § *p* < 0.05 vs. sensitive subclones.

**Figure 2 cells-13-00368-f002:**
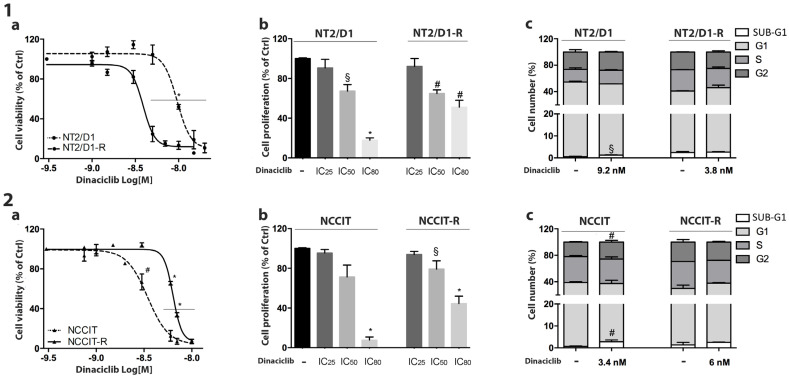
Effect of dinaciclib on NT2/D1/-R and NCCIT/-R cells. (**a**) Cell viability. NT2/D1/-R (1) and NCCIT/-R (2) cells were treated with increasing concentrations of dinaciclib for forty-eight hours as described in Methods. Cell viability was analyzed using the MTT assay. Results are expressed as the percentage of viable cells vs. untreated cells (ctrl). Data are the mean ± S.E.M. of three experiments performed in triplicate. * *p* < 0.0001 vs. untreated cells, # *p* < 0.001 vs. untreated cells. (**b**) Cell proliferation. NT2/D1/-R (**1b**) and NCCIT/-R (**2b**) were treated for forty-eight hours with the corresponding calculated IC25, IC50 and IC80 concentrations of dinaciclib. Cell proliferation was assessed using the cytometer count. Results are expressed as the percentage of viable cells vs. untreated cells (Ctrl) ± SEM. § *p* < 0.05 vs. untreated cells, # *p* < 0.001 vs. untreated cells, * *p* < 0.0001 vs. untreated cells. (**c**) NT2/D1/-R and NCCIT/-R cell-cycle distribution after drug treatment. NT2/D1/-R (**1c**) and NCCIT/-R (**2c**) were treated with the corresponding IC50 concentration of dinaciclib for forty-eight hours, and the cell-cycle distribution was analyzed. Histograms representing the percentage of cells in each cell-cycle phase are reported. § *p* < 0.05 vs. untreated cells, # *p* < 0.001 vs. untreated cells.

**Figure 3 cells-13-00368-f003:**
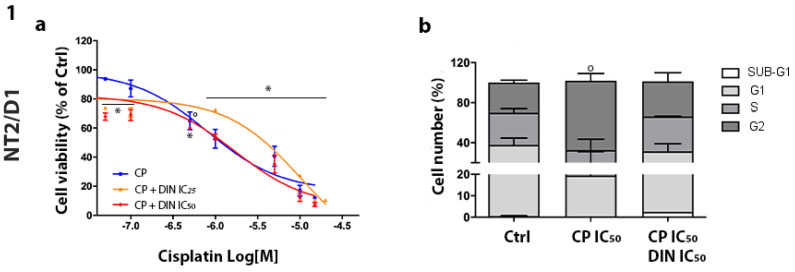
Effect of CP plus dinaciclib treatment on NT2/D1/-R cells. (**a**) Cell viability. NT2/D1 (**1a**) and NT2/D1-R (**2a**) cells were treated with increasing concentrations of CP alone or in the presence of the corresponding IC_25_/IC_50_ dose of dinaciclib (DIN). After forty-eight hours of treatment, cell viability was measured using the MTT assay. Results are expressed as the percentage of viable cells vs. untreated cells (Ctrl) ± SEM. ° *p* < 0.05 vs. untreated cells, § *p* < 0.01 vs. untreated cells; # *p* < 0.001 vs. untreated cells, * *p* < 0.0001 vs. untreated cells. (**b**) Cell-cycle analysis. NT2/D1 (**1b**) and NT2/D1-R cells (**2b**) were treated with the corresponding IC50 concentration of CP/CP plus dinaciclib for forty-eight hours, and the cell-cycle distribution was analyzed. Histograms representing the percentage of cells in each cell-cycle phase are reported. # *p* < 0.001 vs. untreated cells, * *p* < 0.0001 vs. untreated cells.

**Figure 4 cells-13-00368-f004:**
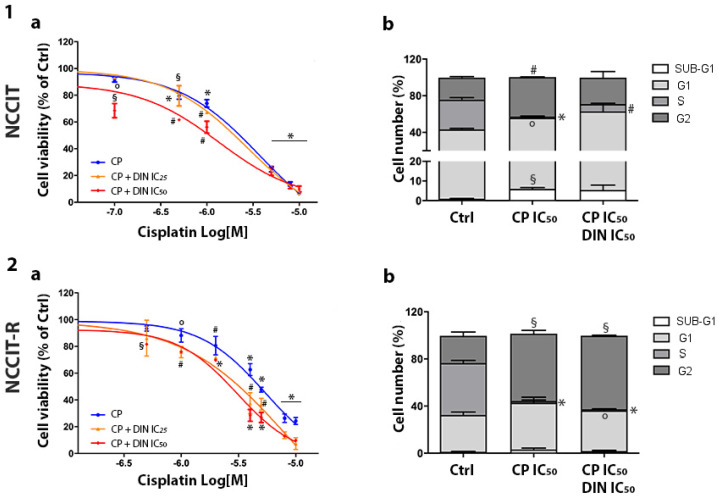
Effect of CP plus dinaciclib treatment on NCCIT/-R cells. (**a**) Cell viability. NCCIT (**1a**) and NCCIT-R (**2a**) cells were treated with increasing concentrations of CP alone or in the presence of the corresponding IC_25_/IC_50_ dose of dinaciclib (DIN). After forty-eight hours of treatment, cell viability was measured using the MTT assay. Results are expressed as the percentage of viable cells vs. untreated cells (Ctrl) ± SEM. ° *p* < 0.05 vs. untreated cells, § *p* < 0.01 vs. untreated cells; # *p* < 0.001 vs. untreated cells, * *p* < 0.0001 vs. untreated cells. (**b**) Cell-cycle analysis. NCCIT (**1b**) and NCCIT-R cells (**2b**) were treated with the corresponding IC50 concentration of CP/CP plus dinaciclib for forty-eight hours, and the cell-cycle distribution was analyzed. Histograms representing the percentage of cells in each cell-cycle phase are reported. ° *p* < 0.05 vs. untreated cells, # *p* < 0.001 vs. untreated cells, * *p* < 0.0001 vs. untreated cells.

**Figure 5 cells-13-00368-f005:**
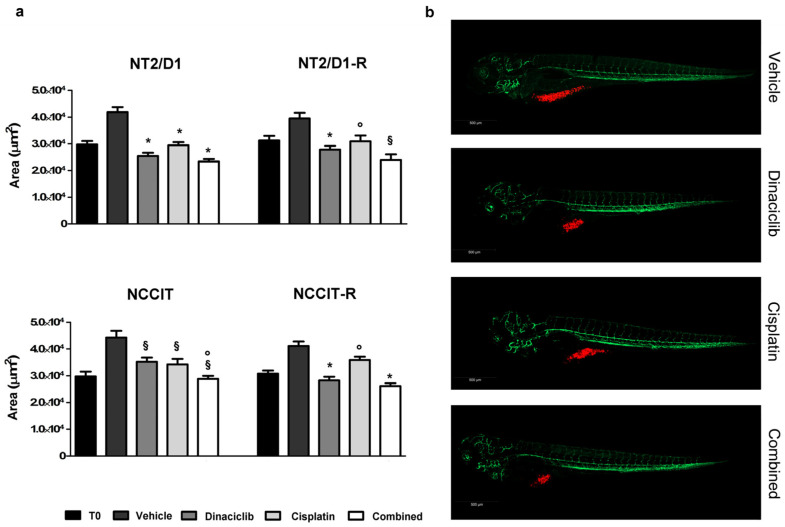
Area of NT2/D1/-R and NCCIT/-R tumor xenograft in AB zebrafish embryos exposed to dinaciclib alone or combined with CP. (**a**) Tumor areas at 120 hpf (T3—end of treatment) of drug-treated and vehicle-treated embryos were measured using Zen 2.3 Black software from ZEISS. Results are reported as area (µm^2^) ± SEM. * *p* < 0.0001, § *p* < 0.01, ° *p* < 0.05 vs. vehicle or single drug as indicated in Table 3. (**b**) Representative, lateral-view pictures of Tg (kdrl:EGFP) untreated and treated embryos at 120 hpf xenografted with NT2/D1 cells. Cells were labeled with a red fluorescent lipophilic dye, while the embryo endothelium was labeled with a green fluorescent protein reporter driven by the kdrl promoter. Images were acquired using a Zeiss LSM 900 confocal microscope at 10× magnification.

**Table 1 cells-13-00368-t001:** CP and dinaciclib IC_50_ values in nonseminoma cell lines.

Cell Line	CP	Dinaciclib
NT2/D1	0.7 µM (95% CI: 0.35–1.43)	9.2 nM (95% CI: 8.3–10.1)
NT2/D1-R	6.1 µM (95% CI: 2.3–16.0)	3.8 nM (95% CI: 3.3–4.5)
NCCIT	2.7 µM (95% CI: 1.1–6.7)	3.4 nM (95% CI: 2.9–4.0)
NCCIT-R	4.1 µM (95% CI: 2.1–8.7)	6.0 nM (95% CI: 5.3–6.8)

**Table 2 cells-13-00368-t002:** CP efficacy when combined with dinaciclib in nonseminoma cell lines.

Cell Line	CP	CP + Dinaciclib IC_25_	CP + Dinaciclib IC_50_
NT2/D1	87.81% ± 0.27%	89.47% ± 1.22%	92.53% ± 1.99%
NT2/D1-R	37.90% ± 3.61%	56.86% ± 0.19% (°)	51.42% ± 1.63% (°)
NCCIT	93.51% ± 2.09%	93.26% ± 0.34%	90.06% ± 3.07%
NCCIT-R	75.70% ± 8.28%	92.63% ± 6.29% (°)	90.11% ± 1.48% (°)

Results are reported as mean ± SEM. ° *p* < 0.05 vs. CP efficacy.

**Table 3 cells-13-00368-t003:** Measured area values of xenograft after vehicle/dinaciclib zebrafish embryo treatment.

Cell Line	Vehicle (T3)	Dinaciclib (T3)	Cisplatin (T3)	Combined (T3)
NT2/D1	39,426 (35,902–43,297)	28,086 (25,649–30,754) *p* < 0.01 vs vehicle	23,597 (21,741–25,612) *p* < 0.01 vs vehicle	22,192 (20,238–24,334) *p* < 0.01 vs vehicle *p* < 0.01 vs cisplatin
NT2/D1-R	36,301 (32,866–40,096)	30,668 (27,332–34,411) *p* < 0.01 vs vehicle	25,436 (23,305–27,762) *p*= 0.030 vs vehicle	19,732 (17,814–21,856) *p* < 0.01 vs vehicle *p* < 0.01 vs cisplatin
NCCIT	41,991 (37,004–47,651)	32,358 (28,149–37,196) *p*= 0.023 vs vehicle	34,113 (30,061–38,711) *p* < 0.01 vs vehicle	28,004 (24,678–31,778) *p* < 0.01 vs vehicle *p* = 0.132 vs cisplatin
NCCIT-R	41,494 (37,533–45,873)	35,879 (32,789–39,261) *p* < 0.01 vs vehicle	25,960 (23,965–28,122) *p* = 0.035 vs vehicle	24,588 (22,455–26,924) *p* < 0.01 vs vehicle *p* < 0.01 vs cisplatin

Results are reported as area in µm^2^ (95% CI).

## Data Availability

The data presented in this study are available on request from the corresponding author.

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
