# Peer review of "The CDK Inhibitor Dinaciclib Improves Cisplatin Response in Nonseminomatous Testicular Cancer: A Preclinical Study"

_cells, 2024, doi:10.3390/cells13050368_

Round 1
Reviewer 1 Report
Comments and Suggestions for Authors
In this manuscript, Rossini and colleagues test the therapeutic effects of a pan-CDK inhibitor, dinaciclib, on the sensitivity of cisplatin-resistant embryonal carcinoma cell lines. This is an important goal as relapse/refractory germ cell tumors (GCT) remain a significant clinical challenge for which few effective treatment options exist. The authors provide interesting new data indicating that two different cisplatin-resistant EC lines show significant up-regulation of CDK5. The authors go on to measure sensitivity of the cells to dinaciclib alone or in combination with cisplatin, both in vitro and in a zebrafish xenotransplantation model in vivo. The results suggest that the combination treatment has some efficacy in both parental and cisplatin-resistant cell lines, although the conclusions are limited by differential dosing (in vitro) and the statistical analyses performed. Overall, the work is important in assessing a new approach for the treatment of cisplatin-resistant GCT and will be of interest to the scientific community, especially after several comments are addressed as noted below:
The authors state (line 170) that CDK1 was the most abundantly expressed CDK in each cell model, but this conclusion cannot be drawn for the data provided. Since the abundance of each CDK is measured with a different antibody (with different affinities for their respective antigens), it is not appropriate to compare the strength of signal between antibodies. Comparing levels of the different CDKs would require quantification of absolute protein abundance based on a known amount of antigen, which was not done here.
The authors appear to misuse the term ‘G0 phase of the cell cycle’. The G0 phase refers to exit of cells from the cell cycle with a diploid DNA content that is indistinguishable from G1 cell cycle phase DNA content. Here the authors appear to use the term to refer apoptotic cells that have less than G1 DNA content, which are typically referred to as ‘sub-G1’ cells rather than G0 cells. This should be corrected throughout. In addition, the authors should provide images of their flow cytometry data used to quantify cell cycle stages as a supplementary figure.
When measuring the combined effects of cisplatin and dinaciclib on parental and cisplatin-resistant cell lines in vitro, the authors combine different doses of dinaciclib (based on IC25 and IC50 calculations) with cisplatin and conclude that the combination is more efficacious in the resistant lines. However, it is not appropriate to compare between sensitive and resistant lines with this approach since different amounts of dinaciclib are used. For NCCIT cells in particular, the resistant lines were treated with almost twice as much dinaciclib than the parental cells. To say that the resistant lines are more responsive to the combination, it seems that parental and resistant lines would need to receive the same amount of dinaciclib and still show the differential sensitivity. For the in vivo experiment, the same amounts of drugs are used for both parental and resistant grafts.
For the analysis of dinaciclib-induced apoptosis (Fig S2) the authors should also report and show the results for the effects on cisplatin-resistant cell lines (in addition to the cisplatin-sensitive parental cells that are shown).
The results section has little to no description of the actual results of the in vivo treatment study. Surprisingly, in vivo treatment with cisplatin seems to have similar effects in the parental and cisplatin-resistant EC xenografts (Fig. 5). Were there any statistically significant differences in how the parental and cisplatin resistant grafts respond to either the single drugs or the combination? These results should be described more clearly (including the apparent lack of cisplatin resistance by the resistant lines in vivo.
For the in vivo treatment experiments, the authors should also perform additional statistical analysis specifically to assess drug interactions. The authors state in the methods section that they did simple t-test or ANOVA analyses for all experiments (and don’t indicate which for these particular experiments). A more thorough statistical assessment would allow determination of whether the combination effect is simply additive or synergistic.
Additional minor comments:
The manuscript (Fig 1 legend for example) refers to NCCIT as a testicular embryonal carcinoma cell line, but these cells are derived from a mediastinal GCT, not a testicular GCT.
The authors indicate that ‘no toxic or side effects were observed even at the highest dose’ of in vivo treatment in zebrafish (line 306). The authors should describe how toxicity was evaluated.
Comments on the Quality of English LanguageThere are some grammar issues to correct but overall the paper is well written.
Reviewer 2 Report
Comments and Suggestions for Authors
Manuscript Number: cells-2798011
Journal Title: Cells (ISSN 2073-4409)
Title: The Cdks-inhibitor dinaciclib improves cisplatin response of non-seminomatous testicular cancer: a preclinical study.
Article Type: Regular Paper
Corresponding Authors: Mariangela Tamburello
All Authors: Elisa Rossini, Mariangela Tamburello, Andrea Abate, Silvia Zini, Giovanni Ribaudo, Alessandra Gianoncelli, Francesca Valcamonico, Nazareno Roberto Suardi, Giuseppe Mirabella, Alfredo Berruti, Sandra Sigala
In this study, the authors have shown the relationship between sensitivity against dinaciclib and the level of Cdk5 expression using non-seminoma cell models. The authors found higher Cdk5 expression levels in CP-resistant models. The double treatment using cisplatin (CP) and dinaciclib results in the elevation of dinaciclib efficacy. The authors concluded that when combined with CP, dinaciclib could help improve non-seminoma TC response to CP.
The experimental works are nicely performed, and the results justify and state the conclusions. Still, unfortunately, this work will not provide an advancement of the current knowledge for this journal's readers. Similar reports have been published somewhere.
I do not recommend this paper for the "Cells" journal.
Comments on the Quality of English LanguageThere are many errors (spacing, size, etc.), and this article does not satisfy the journal guidelines for publishing.
Reviewer 3 Report
Comments and Suggestions for Authors
Please see my comments and suggestions in the document attached

Round 2
Reviewer 1 Report
Comments and Suggestions for Authors
Overall, the authors have responded to my comments effectively and have significantly improved the paper. There are a copy of minor points remaining:
The authors have removed reference to G0 stage of the cell cycle in the text and now more accurately refer to sub-G0 when describing their results. However, figures 2-4 still use G0 in the figure keys. These figures should be updated.
The authors have added Supplemental Figure 4 to show examples of cell cycle profile. I don't see that this new figure is referenced in the main manuscript.
There are a few typos in the newly added text. For example, ‘interestingly’ is spelled incorrectly in line 466. There are few other places where some copy editing is needed.
Comments on the Quality of English LanguageThe authors have significantly improved the writing. There are a few minor issues with newly added text that could be corrected.
Author Response
We modified the text accordingly to all the reviewer comments. The Figure 2, 3 and 4 were updated replacing G0/G1 with SUB-G1.
The Supplemental Figure 4 was mentioned in the main Ms (line 261) end the typos were corrected.
The reviewer comments and suggestion have greatly enhanced the quality of our manuscript, and we are grateful for his/her contribution.
Reviewer 2 Report
Comments and Suggestions for Authors
The authors have made improvements to the manuscript accordingly. It is easier for readers to understand the significance. I recommend this article submitted by Rossini E et al. The pointed details and importance have improved clearly. No further suggestions are made.
Author Response
We are pleased to have addressed the reviewers' requests and improved the quality of our manuscript. We would like to extend our gratitude to the reviewer for his/her support.